# Evaluation of Drug Interactions in Patients Treated with DAAs for Hepatitis C Therapy with Comorbidities and Cardiovascular Issues—A Delphi Consensus Project

**DOI:** 10.3390/jcm11236946

**Published:** 2022-11-25

**Authors:** Claudio Borghi, Alessia Ciancio, Ivan Gentile, Pasquale Perrone Filardi, Patrizio Pasqualetti, Stefano Brillanti

**Affiliations:** 1Internal Medicine, Department of Medical and Surgical Sciences, Policlinic S. Orsola-Malpighi, University of Bologna, 40126 Bologna, Italy; 2Gastroenterology Unit, Department of Internal Medicine, Città della Salute e delle Scienza di Torino (Molinette), University of Turin, 10126 Turin, Italy; 3Department of Clinical Medicine and Surgery, University of Naples Federico II, 80138 Naples, Italy; 4Department of Advanced Biomedical Sciences, AOU Federico II, 80131 Naples, Italy; 5Section of Medical Statistics, Department of Public Health and Infectious Disease, Sapienza Rome University, 00185 Rome, Italy; 6Department of Medicine, Surgery and Neuroscience, University of Siena, 53100 Siena, Italy

**Keywords:** HCV infection, DAAs—direct-acting antivirals, DDIs—drug–drug interactions, Delphi method

## Abstract

Orally administered direct-acting antivirals (DAAs) have dramatically changed the possibility of curing HCV (hepatitis C virus) infection, with the two principal HCV regimens based on the combination of glecaprevir + pibrentasvir (GLE-PIB) and sofosbuvir + velpatasvir (SOF-VEL). A combination of drugs containing NS3/4A protease inhibitors, as well as the fact that almost all HCV patients can be treated at present, may expose patients to a higher rate of drug–drug interactions (DDIs). The hepatitis C treatment recommendations from the EASL (European Association for the Study of the Liver) state that, prior to starting treatment with a DAA, a detailed drug history should be taken; yet, the decision on managing the potential DDIs is not always clear. For this reason, a group of Italian cardiologists and hepatologists promoted a survey among colleagues to assess the controversial issues when treating patients with chronic hepatitis C taking concomitant cardiovascular drugs, aiming to reach a consensus on the best practice to apply when treating a patient with chronic hepatitis C who is taking concomitant drugs for cardiovascular diseases. Two consecutive questionnaires were proposed between June and July 2022 to a qualitative Expert Panel (EP) of 14 gastroenterologists, infectologists, hepatologists, and internists, with statistical analyses performed on 100% of the responses for both questionnaires. Agreement among experts was assessed following the Delphi method as developed by the RAND Corporation. The interviewed experts consider DDIs a critical clinical problem to be evaluated in HCV patients. Therefore, dose changes, drug substitution, and discontinuation of concomitant cardiovascular drugs should be discouraged, even if planned for a relatively short period. Since oral DAAs have different DDIs profiles, hepatologists should prefer the antiviral DAA combination presenting the lowest instance of potential interactions.

## 1. Introduction

Hepatitis C is the most common cause of chronic liver disease and the most frequent indication for liver transplantation in Western countries. In 2015, the World Health Organization (WHO) estimated that approximately 100 million people globally had serologic evidence of HCV (hepatitis C virus) exposure, and 71 million people had chronic HCV infection (prevalence of 1%) [1].

During the last decade, the new generation of oral direct-acting antivirals (DAAs) has dramatically changed the possibility of curing HCV infection. Current DAAs can achieve a sustained virological response (SVR) in more than 98% of patients, and have a good safety profile [2].

Oral DAAs are administered as therapeutic combination regimens of additional agents, which are administered as a fixed combination. These regimens are easy to administer, well tolerated, and highly effective across a broad range of patient populations, including older patients and patients previously considered “difficult-to-treat”.

The two principal HCV regimens are based on the combination of glecaprevir + pibrentasvir (GLE-PIB) and sofosbuvir + velpatasvir (SOF-VEL). Glecaprevir is an NS3/4A serine protease inhibitor. Sofosbuvir is an NS5B nucleotide analogue inhibitor of the NS5B polymerase. Pibrentasvir and velpatasvir are NS5A inhibitors. A combination of drugs containing NS3/4A protease inhibitors may expose patients to a higher rate of drug–drug interactions (DDIs) [3].

As previously stated, since almost all HCV patients can now be treated, it results in the prevalence of more comorbidities and multiple concomitant medications, putting a significant proportion of patients at risk of clinically significant drug–drug interactions (DDIs). Real-world studies have shown that more than 50–70% of patients were taking concomitant medications, and the prevalence of potentially significant DDIs was present in as high as 40% of patients [4,5].

In the general senior population, concomitant medications potentially at risk of significant DDIs are mainly those prescribed for cardiovascular problems, especially lipid-lowering agents and antihypertensive drugs [6].

Cardiovascular diseases—or conditions often preceding the diagnosis of HCV infection—and patients evaluated by hepatologists are taking the needed drugs for months or years. Therefore, hepatologists are often required to ascertain and decide whether the concomitant therapy may interact with DAAs, and in which form and level of interaction (potential weak, potential, and contraindication).

The recommendations for treatment of hepatitis C from the European Association for the Study of the Liver (EASL) state that, prior to starting treatment with a DAA, a complete and detailed drug history should be conducted, including all prescribed medications, over-the-counter drugs, herbal and vitamin preparations, and any illicit drug use discussed and documented [2]. A pre-treatment appointment should be considered to rationalize prescribing. The pharmacokinetic profiles and how HCV drugs impact drug–drug interactions should be evaluated, and the consulting of the website www.hep-druginteractions.org (accessed on 30 June 2022) with a list of over 800 comedications is recommended before starting HCV treatment.

Even if the picture seems well defined, the decision on managing the potential DDIs is not always clear. For example, a consensus on whether to reduce, stop, or substitute a concomitant drug for a cardiovascular condition or select a DAA upon another is often a personal decision.

Thus, taking into account the widening of the HCV population that can be treated and the increase in the number of individuals with concomitant diseases and treatments—and, consequently, the possibility of encountering situations of DDIs in clinical practice that are not precisely regulated by guidelines and recommendations—a group of Italian cardiologists, hepatologists, and infectologists promoted a survey among expert colleagues to assess the level of consensus regarding controversial issues and the best practice to apply when treating a patient with chronic hepatitis C who is taking concomitant drugs for cardiovascular diseases. The degree of agreement among participants and the conclusions were drawn using the Delphi methodology [7].

## 2. Subjects and Methods

This study was undertaken between June and July 2022. The topics covered by the survey were developed by the study authors and presented to a panel of responding clinicians in the form of two consecutive questionnaires (Appendix A) consisting of variously structured questions. The greater part of these involved a response based on a scale ranging from 1 (maximum disagreement) to 9 (maximum agreement), while in only one of the questions, this scale ranged from 1 to 4. In the other two questions, it was possible to select multiple choices, and in one, the expected responses included three possible answers.

To initiate the study, an Advisory Board of the six authors—considered experts in this area—was convened, and based on a thorough review of the literature, prepared a first questionnaire (Q1) that included questions on physicians’ demographics, type of institutions where they were working, years of experience in the field, and geographic area of their facilities. Some of these data were used as covariates (respondents’ age, years of involvement in the management of cardiovascular disease, and scope of their work). In addition, centers involved were classified with respect to their characteristics. The Advisory Board then met to discuss the results of the Q1 survey and developed a second questionnaire (Q2) to resolve issues that were not clear from Q1, and to better assess the appropriateness of some of the diagnostic/therapeutic/management procedures.

The two questionnaires were qualitatively validated in terms of content and face validity. In detail, content validity (relevance and comprehensiveness of the items chosen to explore drug interactions in patients treated with DAAs for hepatitis C therapy) was assessed within the Advisory Board, while face validity was qualitatively tested by colleagues of the members of the Advisory Board who provided suggestions about understandability and ambiguities before obtaining the final versions of the questionnaires.

Q1 and Q2, prepared as computerized questionnaires (compilable PDF, 1984–2022 Adobe), sent and returned by e-mail, were administered to a qualitative Expert Panel (EP) consisting of gastroenterologists, infectologists, hepatologists, and internists comprising 14 Italian centers (including those of the Advisory Board) (Appendix A) selected randomly and distributed throughout the country. Q1 was sent out to the EP in June 2022, with statistical analysis performed on the responses of all participants. Q2, allowing the refinement of some topics that had generated ambiguous responses in Q1, was sent in July 2022 and responses were collected by the same month. Data were analysed by means of Excel (Microsoft Office), developing specific and appropriate functions for the objectives of this study (see next paragraph). Statistical analyses were performed on 100% of the respondents both for Q1 and Q2.

The agreement among experts was further assessed following the Delphi method as developed by the RAND Corporation [7]. Delphi is a well-established and validated consensus-building process for developing agreement and making group-based decisions in various fields [8,9,10,11,12]. Traditionally based on the three core concepts of anonymity, controlled feedback, and statistical group response, the method is routinely used in health research and clinical challenges [11]. The method requires the use of a scale ranging from one (maximal disagreement) to nine (maximal agreement), with five corresponding to a neutral opinion about any given item. Scores provided by the experts were then statistically elaborated to obtain an appropriate “index of consensus”. According to the RAND/UCLA Appropriateness Method User’s Manual, Inter-Percentile Range Adjusted for Symmetry (IPRAS) scores, which are a measure of score dispersion adjusted for panel symmetry, were used to determine the level of agreement for each item. The rationale is that, when ratings are symmetric, the Inter-Percentile Range (IPR) required to label an indication as disagreement is smaller than when they are asymmetric. Asymmetry was defined as “the distance between the central point of the IPR and the central point of the 1–9 scale, i.e., 5”. Since the more asymmetric the ratings are, the larger the IPR required to say that there is disagreement, the following mathematical function was developed: IPRAS = IPRr + (AI × CFA), where IPRr is the IPR required for disagreement when perfect symmetry exists; AI is the asymmetry index; CFA is the correction factor for asymmetry. A statement or indication is rated with a disagreement if IPRi > IPRASi. Based on IPR and IPRAS computation, it is possible to classify each statement with the appropriateness of a given diagnostic/therapeutic strategy in the following categories: appropriate (panel median of 7–9, without disagreement); uncertain (panel median of 4–6 or any median with disagreement); inappropriate (panel median of 1–3, without disagreement).

## 3. Results

### Study Population

Seventeen experts were invited to participate in the projects. Three of them did not answer in the due time and, thus, statistical analysis was centered around data provided by fourteen participants who answered to both Q1 and Q2 questionnaires without any drop-out. The median age of the respondents was 61 years (min = 41, max = 71). Detailed characteristics of respondents are presented in Table 1.

Participants were asked to rate the tools used to assess the risk of drug interactions before starting HCV therapy, with the highest score obtained by the University of Liverpool database (median = 9), followed by personal experience (7), leaflet (6.5), EASL recommendations (5), and Medscape (1).

The most common comorbidities of the participants’ HCV patients were hypertension (*n* = 12), dyslipidemia (*n* = 11), atrial fibrillation (*n* = 8), anxiety (*n* = 8), and cardiopathy (*n* = 7); this picture was confirmed by the most common drugs (anti-hypertensive, lipid-lowering agents, PPI inhibitors, and anxiolytics).

As expected, respondents showed strong consensus with the statement “HCV infection is a risk factor for the development of cardiovascular disease” (median = 8, min = 6, max = 9), and for the statement “Elimination of viral infection with direct antiviral drugs reduces the risk of development or progression of cardiovascular diseases” (median = 8, min = 5, max = 9). On the other hand, no clear consensus emerged—even if the sequence B > A = C > D appeared the most relatively appropriate—when participants were asked to determine the best sequence ranking the following items: “A. Consider the CV effects of HCV treatment”; “B. Classification of the CV risk profile”; “C. Consider drug treatment of CVD compatible and with fewer drug interactions with HCV therapy”; “D. Shared management between cardiologist and hepatologist/infectious disease specialist”. 

Table 2 and Figure 1 show the indexes of appropriateness evaluated according to the RAND/UCLA Method. Regarding the first questionnaire, the experts agreed on the correctness/appropriateness of the majority of statements. Moreover, they agreed on the inappropriateness of the following three statements: “21. The profile of drug interactions between HCV antiviral drugs and the main statins used is similar”; “23. Discontinuation of statin use for 8–12 weeks can also be tolerated in secondary prevention”; “24. Management of drug interactions with HCV antiviral drugs is simple and does not require special multidisciplinary approaches”. For two statements, the combined evaluation of median consensus score and of the inter-rater variability indicated a lack of agreement (“uncertainty”) with the following statements: “16. For the management of hypertension in the HCV patient, Angiotensin-converting enzyme (ACE) inhibitors should be given only as needed and preferably in combination with sofosbuvir/velpatasvir (SOF/VEL) and not in combination with glecaprevir/pibrentasvir (GLE/PIB)”; “22. Discontinuation of statin use for 8–12 weeks does not result in a significant increase in CVD risk in primary prevention”.

After the findings of the first round were presented to participants, the board proposed a focus on and clarification of the “uncertain statements”, aiming to remove possible ambiguities. As shown in Table 2 and in Figure 2, the answers to Q2 resulted in a consensus among experts, who agreed that the statement “E. For the management of hypertension in the HCV patient, ACE inhibitors should be given in combination with glecaprevir/pibrentasvir (GLE/PIB)” was inappropriate, with the other statements being appropriate. 

## 4. Discussion

The results of the current Delphi project can be summarized as follows:Treating HCV patients with comorbidities and comedications requires a particular multidisciplinary approach.Treating HCV patients taking medications for cardiovascular problems requires particular attention to the presence and management of drug–drug interactions.In HCV patients taking antihypertensive, antithrombotic or lipid-lowering agents for the prevention or treatment of cardiovascular diseases, the relevant drugs should not be withdrawn, changed or reduced in dosage—even for a short period—when treating hepatitis C.Oral DAAs have different profiles in terms of drug–drug interactions.The potential for DDIs with HCV DAAs is more frequent with GLE-PIB than SOF-VEL, and DDIs represent a significant reason one regimen may be selected in favor of another.

GLE-PIB and SOF-VEL combination therapies have shown similarly good efficacy and tolerability, allowing more patients with chronic hepatitis C to be treated [13]. Therefore, patients with previous contraindications—such as old age or comorbid diseases—can be effectively treated with all-oral-DAA regimens. For this reason, in clinical practice, many HCV patients take multiple medications when diagnosed with hepatitis C.

Frequent concomitant medications are antihypertensive, antithrombotic, or lipid-lowering agents (i.e., statins) that may interact with prescribed DAAs. Therefore, hepatologists should perform a complete and detailed drug history before starting treatment with a DAA, along with evaluating the pharmacokinetic profiles and how HCV drugs impact drug–drug interactions. The University of Liverpool’s up-to-date website www.hep-druginteractions.org (accessed on 30 June 2022) is a fundamental tool used to assess and check the potential level of drug–drug interactions.

When checking for DDIs between the ongoing therapy for cardiovascular conditions and the DAAs, the hepatologist may find a potentially significant interaction that should be avoided. One option would be to withdraw, change, or reduce the dosage of the concomitant drug when treating hepatitis C due to the relatively short duration of the antiviral therapy. However, experts participating in the current study agreed that this is not the right choice. The evidence informing this consensus relies on multiple results regarding the management of cardiovascular therapy: (a) different drugs may not have the exact therapeutic equivalence [14], (b) reducing the therapeutic dose is associated with unfavorable outcomes [15], and (c) treatment discontinuation may have significant detrimental effects on primary clinical outcomes, even if the drug has been stopped for a short period [16,17].

Another interesting point highlighted by the expert consensus is that, in real-world practice, the temporary discontinuation of cardiovascular therapy for the period of antiviral treatment carries the risk that the patient will prolong the discontinuation beyond the period he or she has been instructed to. 

On the other hand, the choice of prescribing a DAA despite a potentially significant drug interaction cannot be recommended; negative consequences of drug interactions with DAAs may include decreased concentration resulting in loss of efficacy, or, on the contrary, increased levels associated with drug toxicity. For example, drugs used for HCV patients with cardiovascular disease, such as statins, are substrates of various drug transporters and drug-metabolizing enzymes that are inhibited by specific DAAs, resulting in a clinically relevant increase in statin plasma concentrations and, consequently, potential safety issues [3,5,18].

Oral DAA combinations differ in composition. In particular, glecaprevir is an NS3/4A serine protease inhibitor, and sofosbuvir is an NS5B nucleotide analogue inhibitor of the NS5B polymerase. Pibrentasvir and velpatasvir are NS5A inhibitors. DDIs have previously been reported as being most frequent in regimens containing NS34A serine protease inhibitors, especially with lipid-lowering agents [3]. In the present survey, the experts agreed that the profile of drug interactions between HCV antiviral drugs and statins is different, since the combination of SOF-VEL has shown less prevalent and significant potential interactions with frequently prescribed statins [19].

Similarly (although to a lesser extent), the SOF-VEL combination has shown less significant interaction with ACE inhibitors (particularly with enalapril), leading experts to prefer this DAA combination for patients taking ACE inhibitors [20].

Regarding the two most commonly prescribed ACE inhibitors (ramipril and enalapril), and based on metabolism and clearance profile, it is unlikely that ramipril have a clinically significant interaction. Ramipril is metabolized by esterases and glucuronidation, and these pathways are not affected by SOF-VEL. Enalapril, a prodrug hydrolyzed in the liver to enalaprilat, is a substrate but not an inhibitor of OATP1B1. Although concentrations may increase due to mild inhibition of OATP1B1 by velpatasvir, this will unlikely have clinical significance. On the other hand, enalapril is a substrate of OATP1B1, and concentrations may increase due to inhibition of OATP1B1 by GLE-PIB.

The Delphi method allowed the experts to reach the final consensus that in the case of polypharmacotherapy, it is important not to change the current cardiovascular therapy by favoring the choice of the antiviral with less interference, and evidence supports that SOF/VEL requires fewer modifications of contraindicated concomitant drugs before starting DAA treatment. This statement contrasts with experiences reported in real-world clinical practice, where dose modification, switch, or interruption of concomitant drugs frequently occur at different rates among DAAs, suggesting a different perception regarding the potential severity of DDIs. In addition, many patients may receive contraindicated cotreatment, although most have a dose decrease [4,5]. The clinical relevance of DDI suggests that awareness of comedication administration should be increased. Attention should be given to major DDIs and their potential adverse outcomes.

## 5. Conclusions

The current Delphi project showed that a contemporary cohort of Italian expert cardiologists and hepatologists consider DDIs a critical clinical problem to be addressed and evaluated in HCV patients taking comedications for cardiovascular issues. Therefore, dose changes, drug substitution, and discontinuation of concomitant cardiovascular drugs should be discouraged, even if planned for a relatively short period. Furthermore, since oral DAAs have different profiles in terms of drug–drug interactions, hepatologists should prefer the antiviral DAA combination presenting the lowest instance of potential interactions.

## Figures and Tables

**Figure 1 jcm-11-06946-f001:**
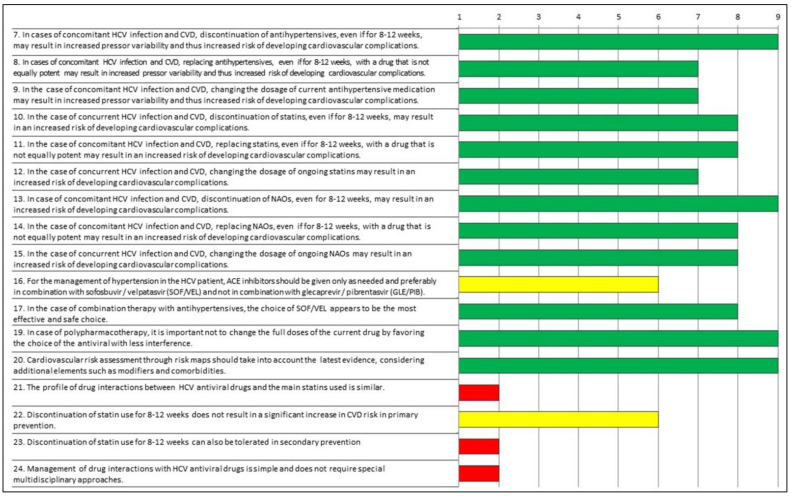
Appropriateness of responses to Questionnaire 1 evaluated according to the RAND/UCLA Method. Green: appropriate; yellow: uncertain; red: inappropriate.

**Figure 2 jcm-11-06946-f002:**
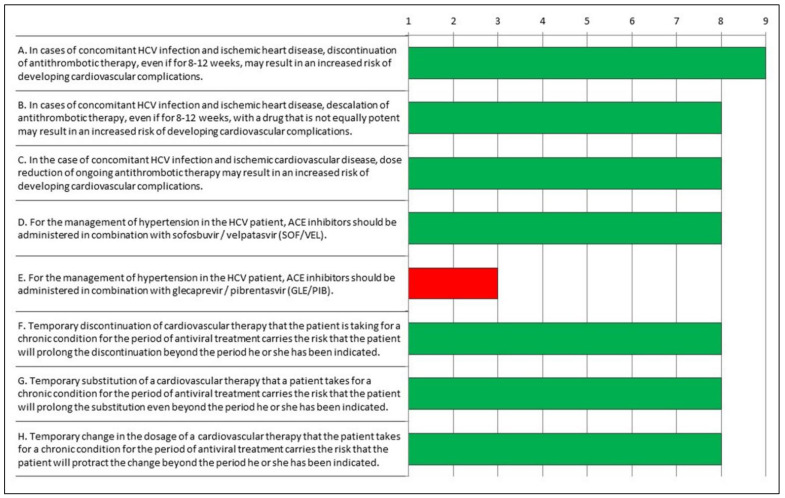
Appropriateness of responses to Questionnaire 2 evaluated according to the RAND/UCLA Method. Green: appropriate; red: inappropriate.

**Table 1 jcm-11-06946-t001:** Characteristics of the study population.

	*n* = 14
Age, years (median, min–max)	61, 41–71
Overall working experience, years (median, min–max)	35, 17–46
Field of prevalent activity	
Cardiology (*n*)	3
Hepatology (*n*)	8
Other (*n*)	3
Experience in the Field of prevalent activity, years (median, min–max)	28.5, 7–35
Geographical area	
North (*n*)	4
Center (*n*)	6
South Islands (*n*)	4

**Table 2 jcm-11-06946-t002:** Appropriateness Indexes evaluated according to the RAND/UCLA Method.

Questionnaire	Item	Median	IQR	IPRAS	Evaluation
Q1	7. In cases of concomitant HCV infection and CVD, discontinuation of antihypertensives, even if for 8–12 weeks, may result in increased pressor variability and thus increased risk of developing cardiovascular complications.	9	1.8	7.5	Appropriate
Q1	8. In cases of concomitant HCV infection and CVD, replacing antihypertensives, even if for 8–12 weeks, with a drug that is not equally potent may result in increased pressor variability and thus increased risk of developing cardiovascular complications.	7	2.0	5.4	Appropriate
Q1	9. In the case of concomitant HCV infection and CVD, changing the dosage of current antihypertensive medication may result in increased pressor variability and thus increased risk of developing cardiovascular complications.	7	2.0	5.4	Appropriate
Q1	10. In the case of concurrent HCV infection and CVD, discontinuation of statins, even if for 8–12 weeks, may result in an increased risk of developing cardiovascular complications.	8	2.8	5.4	Appropriate
Q1	11. In the case of concomitant HCV infection and CVD, replacing statins, even if for 8–12 weeks, with a drug that is not equally potent may result in an increased risk of developing cardiovascular complications.	8	2.0	5.4	Appropriate
Q1	12. In the case of concurrent HCV infection and CVD, changing the dosage of ongoing statins may result in an increased risk of developing cardiovascular complications.	7	2.0	5.4	Appropriate
Q1	13. In case of concomitant HCV infection and CVD, discontinuation of DOACs, even for 8–12 weeks, may result in an increased risk of developing cardiovascular complications.	9	0.8	8.3	Appropriate
Q1	14. In the case of concomitant HCV infection and CVD, replacing DOACs, even if for 8–12 weeks, with a drug that is not equally potent may result in an increased risk of developing cardiovascular complications.	8	1.0	6.1	Appropriate
Q1	15. In the case of concurrent HCV infection and CVD, changing the dosage of ongoing DOACs may result in an increased risk of developing cardiovascular complications.	8	1.8	6.2	Appropriate
Q1	16. For the management of hypertension in the HCV patient, ACE inhibitors should be given only as needed and preferably in combination with sofosbuvir/velpatasvir (SOF/VEL) and not in combination with glecaprevir/pibrentasvir (GLE/PIB).	6	2.8	5.4	Uncertain
Q1	17. In the case of combination therapy with antihypertensives, the choice of SOF/VEL appears to be the most effective and safe choice.	8	1.0	6.1	Appropriate
Q1	19. In case of polypharmacotherapy, it is important not to change the full doses of the current drug by favoring the choice of the antiviral with less interference.	9	0.0	8.4	Appropriate
Q1	20. Cardiovascular risk assessment through risk maps should take into account the latest evidence, considering additional elements such as modifiers and comorbidities.	9	1.0	7.6	Appropriate
Q1	21. The profile of drug interactions between HCV antiviral drugs and the main statins used is similar.	2	4.3	6.6	Inappropriate
Q1	22. Discontinuation of statin use for 8–12 weeks does not result in a significant increase in CVD risk in primary prevention.	6	5.5	2.4	Uncertain
Q1	23. Discontinuation of statin use for 8–12 weeks can also be tolerated in secondary prevention	2	4.3	6.6	Inappropriate
Q1	24. Management of drug interactions with HCV antiviral drugs is simple and does not require special multidisciplinary approaches.	2	2.8	6.8	Inappropriate
Q2	A. In cases of concomitant HCV infection and ischemic heart disease, discontinuation of antithrombotic therapy, even if for 8–12 weeks, may result in an increased risk of developing cardiovascular complications.	9	1.0	7.6	Appropriate
Q2	B. In cases of concomitant HCV infection and ischemic heart disease, descalation of antithrombotic therapy, even if for 8–12 weeks, with a drug that is not equally potent may result in an increased risk of developing cardiovascular complications.	8	1.0	6.1	Appropriate
Q2	C. In the case of concomitant HCV infection and ischemic cardiovascular disease, dose reduction of ongoing antithrombotic therapy may result in an increased risk of developing cardiovascular complications.	8	1.8	6.0	Appropriate
Q2	D. For the management of hypertension in the HCV patient, ACE inhibitors should be administered in combination with sofosbuvir/velpatasvir (SOF/VEL).	8	3.0	6.1	Appropriate
Q2	E. For the management of hypertension in the HCV patient, ACE inhibitors should be administered in combination with glecaprevir/pibrentasvir (GLE/PIB).	3	3.8	4.5	Inappropriate
Q2	F. Temporary discontinuation of cardiovascular therapy that the patient is taking for a chronic condition for the period of antiviral treatment carries the risk that the patient will prolong the discontinuation beyond the period he or she has been indicated.	8	1.8	6.2	Appropriate
Q2	G. Temporary substitution of a cardiovascular therapy that a patient takes for a chronic condition for the period of antiviral treatment carries the risk that the patient will prolong the substitution even beyond the period he or she has been indicated.	8	2.5	6.0	Appropriate
Q2	H. Temporary change in the dosage of a cardiovascular therapy that the patient takes for a chronic condition for the period of antiviral treatment carries the risk that the patient will protract the change beyond the period he or she has been indicated.	8	2.5	6.0	Appropriate

IQR: interquartile range; IPRAS: Inter-Percentile Range Adjusted for Symmetry; DOAC: direct-acting oral anti-coagulants.

## Data Availability

Data will be made available to other investigators upon a reasonable request.

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
