# Peer review of "Evaluation of Drug Interactions in Patients Treated with DAAs for Hepatitis C Therapy with Comorbidities and Cardiovascular Issues—A Delphi Consensus Project"

_jcm, 2022, doi:10.3390/jcm11236946_

Round 1
Reviewer 1 Report
The manuscript aims to assess the grade of consensus about controversial issues and the best practice to apply when treating a patient with chronic hepatitis C who is taking concomitant drugs for cardiovascular diseases. The authors have focused on questions about drug-drug interactions and used two questionaries with specific questions for 14 gastroenterologists, infectivologists, and hepatologists. Some points should be clarified.
It is a qualitative study, but the authors should describe the percentage of 14 participants.
The author should relate the validation part of two questionaries.
Also, I don’t know, but Supplementary material 3 is the same as table 2 in of manuscript.
Finally, in Introduction section is possible detect one paragraph similar citation with other authors. 12% of plagirism were detected in manuscript.
"The recommendations for treatment of hepatitis C from the European... 800 comedications is recommended before starting HCV treatment."
Reference:
Kondili LA, Aghemo A, Andreoni M. Challenges on the achievement of World Health Organization goals for HCV elimination in Italy: need for a Regional programmatic approach on screening and linkage to care. Commentary. Ann Ist Super Sanita. 2021 Jul-Sep;57(3):201-204. doi: 10.4415/ANN_21_03_02. PMID: 34554113.
Reviewer 2 Report
I appreciate the opportunity to review this interesting research manuscript Title: Evaluation of drug interactions in patients treated with DAAs for hepatitis C therapy with comorbidities and cardiovascular issues - A Delphi Consensus Project.
In this study, authors demonstrated the Evaluation of drug interactions in HCV patients treated with DAAs. The experiment design and the presentation of data are good. The manuscript was well-analyzed, and I am satisfied with the figures. There is a clear effort in the work. However, several points should be addressed by the authors. This research might have its clinical significance but still, improvements are required. The recommendations for the manuscript are explained below.
Comments 1 The authors should define the abbreviations for the first time in the abstract, text of the manuscript, and figure legends, and then follow with the abbreviations in the whole of the manuscript.
Comments 2 In my opinion, the introduction section could not show the gap in science well in this research. The authors need to improve it.
Comments 3 Please go over your manuscript text and ensure it is written in an acceptable English language.
Comments 4 Authors must clear the problem statement and target in the abstract
Comments 5 Introduction needs clear and well defined objectives of the study
Comments 6 Why the author selects the Delphi method of consensus
Comments 7 Do the authors have a special reason for choosing the designing consensus on HCV therapies in Italy
Comments 8 Why the combination of SOF/VEL need modifications before starting as DAA treatment.
Comments 9 Authors must explain the best Oral DAAs which have best effect profiles in terms of drug-drug interactions point and recommended by all seven participants
Comments 10 In the result section, the authors should define the abbreviations of each term.
Comments 11 authors must explain the effect of GLE-PIB and SOF-VEL combination therapies on acute and chronic HCV
Comments 12 Why the SOF-VEL combination has shown less significant interaction with ACE-inhibitors
Comments 13 The quality of some pictures is poor. Some word is hard to recognize. The authors need to revise that.
Comments 14 It is suggested to check the manuscript for English grammar once more. Also, the authors have written very short sentences or non-academic sentences in the manuscript; therefore, they need to correct and edit with professional English editing.
